# Creating Anthropomorphic Phantoms via Unsupervised Convolutional Neural Networks

**Junyu Chen**[1,2]                                                                    JCHEN245@JHMI.EDU

**Ye Li**[1,2]                                                                              YLI192@JHU.EDU

**Yong Du**[1]                                                                            DUYONG@JHU.EDU

**Eric C. Frey**[1,2]                                                                    EFREY@JHMI.EDU

[1] *Department of Radiology and Radiological Science, Johns Hopkins Medical Institutes, USA*

[2] *Department of Electrical and Computer Engineering, Johns Hopkins University, USA*

## Abstract

Computerized phantoms play an important role in medical imaging research. They can serve as a gold standard for evaluating and optimizing medical imaging analysis, processing, and reconstruction methods. Existing computerized phantoms model anatomical variations through organ and phantom scaling, which does not fully capture the range of anatomical variations seen in humans. Here, we present a registration-based method for creating highly realistic and detailed anthropomorphic phantoms. The proposed registration method is built on the use of an unsupervised convolutional neural network (ConvNet) that warps the four-dimensional Xtended Cardiac-Torso (XCAT) phantom to a patient CT scan. The registration ConvNet iteratively optimizes an SSIM-based loss function for a given image pair without prior training. We experimentally show substantially improved image similarity of the generated phantom using the proposed method to a patient image.

**Keywords:** Computerized Phantoms, Image Registration, Convolutional Neural Networks

## 1. Introduction

Computerized phantoms are widely used in many medical imaging applications. The four-dimensional Xtended Cardiac-Torso (XCAT) phantom (Segars et al., 2010) was developed to provide highly realistic and detailed anatomical models with known structural and physiological properties. The XCAT phantom provides a parameterized model to create anatomical variations through scaling tissue volumes. However, this simple scaling of organ shapes does not fully capture the anatomical variations seen in humans. Existing methods (Segars et al., 2013) have been proposed to create anatomically realistic phantoms via deformable image registration (DIR). They register phantom labels to the label maps of clinical CT images using DIR; the resulting deformation fields were then applied to the phantom, creating new phantoms that capture the patients' anatomical shapes. However, a downside of such methods is that they heavily rely on the segmentation of multiple organs in the clinical CT images, which can be time-consuming to generate. In this work, we present a novel approach to create extremely detailed computerized phantoms using an unsupervised ConvNet-based DIR. Specifically, we treat the ConvNet as an optimization tool that iteratively minimizes a loss function for an image pair comprised of the XCAT phantom and a patient CT. A new ConvNet is initialized for each image pair, thus the proposed method does not require training and it is fully and truly unsupervised. Full details of the proposed method are described in our published paper (Chen et al., 2020).

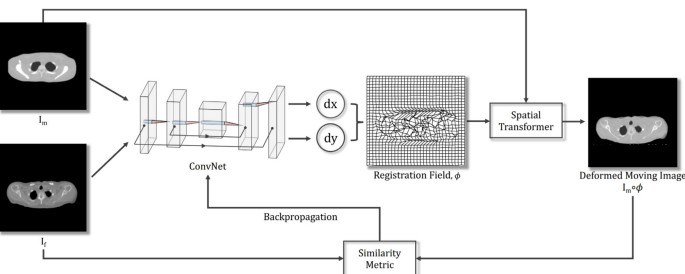

**Figure 1:** Overview of the proposed method.

## 2. Method

The phantom used in this work was created from the 3D attenuation distribution of XCAT phantom (Segars et al., 2010). This single 3D phantom served as the moving image, and it was deformed to multiple patient CT images. Let $I_f$ and $I_m$ be fixed and moving image volumes, that is, a patient CT image and the XCAT phantom. Fig. 1 shows an overview of the proposed method. The ConvNet takes one $I_f$ and one $I_m$ as its inputs. The network learns from a single image pair and produces a deformation field, $\phi$. Then moving image $I_m$ is warped with $\phi$ using a spatial transformer. The loss determined by the image similarity measure between $I_m$ and $I_f$ and the smoothness constraint of $\phi$ is then backpropagated to update the network's parameters. Once the loss converges, the resulting $\phi$ then represents the optimal registration field for the given image pair. Since no aspect of the ConvNet is learned from a prior training stage, the method follows a fully unsupervised paradigm.

**ConvNet Architecture** The ConvNet follows a U-Net-like "hourglass" architecture (Ronneberger et al., 2015). The input to the network is formed by concatenating $I_m$ and $I_f$ into a single volume. The network consists of 19 convolutional layers. The upsampling in the decoder was done by "up-convolution". Each of the upsampled feature maps was concatenated with the corresponding feature map from the encoding path (i.e., skip connections).

**Loss Function** The loss function takes the form of:

$$\mathcal{L}(I_m, I_f, \phi; \theta) = \mathcal{L}_{sim}(I_m \circ \phi, I_f; \theta) + \lambda \mathcal{R}(\phi; \theta), \tag{1}$$

where $\theta$ denotes the set of parameters in the network $f_\theta$, $\phi = f_\theta(I_m, I_f)$, $\mathcal{L}_{sim}$ is the image similarity measure between $I_m$ and $I_f$, and $\mathcal{R}$ denotes the smoothness constraint placed on the deformation field $\phi$. We studied several choices of $\mathcal{L}_{sim}$, including mean squared error (MSE), cross correlation (CC), and mutual information (MI), as well as different choices of $\mathcal{R}$, such as diffusion regularization, total variation regularization, non-negative Jacobian (Kuang and Schmah, 2019), and Gaussian smoothing. This work also presents a novel $\mathcal{L}_{sim}$ that is based on the Structural similarity index (SSIM) (Wang et al., 2004) and Pearson's correlation coefficient (PCC):

$$\mathcal{L}_{sim}(I_m \circ \phi, I_f; \theta) = 0.5 \cdot (1 - \text{SSIM}(I_m \circ \phi, I_f)) + 0.5 \cdot (1 - \text{PCC}(I_m \circ \phi, I_f)), \tag{2}$$

we found that use of SSIM alone made the registration network sensitive to noise or artifacts (i.e., the network tended to model noise in the deformed image), whereas the network trained using PCC alone was robust to noise but did not model image details. Thus we balanced SSIM and PCC with an equal weight of 0.5.

## 3. Results and Conclusions

This work aims at creating anthropomorphic phantoms by registering the XCAT phantom with patient CT scans. The resulting deformation field was then used to deform the

|  | Affine only | SyN (CC) | VoxelMorph (CC) | UnsupConvNet[1] |
|---|---|---|---|---|
| SSIM | 0.83±0.008 | 0.89±0.011 | 0.92±0.006 | 0.96 ± 0.007 |
| MSE | 69.2±2.7 | 52.8±4.1 | 43.5±4.8 | 37.3 ± 5.1 |

**Table 1:** Quantitative comparisons. Qualitative results are shown in (Chen et al., 2020).

XCAT phantom label map. The method was evaluated using 1153 2D-transaxial slices from (Kurdziel et al., 2012). However, the implementation of the method is dimensionally independent. We compare the proposed registration method terms of SSIM and MSE to Symmetric Normalization (SyN) (Avants et al., 2008), and VoxelMorph (Balakrishnan et al., 2018). Table 1 shows the qualitative and quantitative results. Overall, the method provided better XCAT-to-CT mapping. Despite the fact that the proposed method without any regularization produced a deformed phantom image that was almost identical in appearance to the target CT image, the warped label maps might not be realistic. This was caused by the nonsmooth deformation field and by the different interpolation methods used for warping the phantom images and the label maps (i.e., bi-cubic for phantom images and nearest neighbor for label maps). Therefore, adding a regularizer to the loss function enforced the smoothness in the deformation field and produced more realistic warped label maps. Both quantitative and qualitative analyses indicated that the proposed method provided the best registration results.

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
