# OpenReview forum: "Creating Anthropomorphic Phantoms via Unsupervised Convolutional Neural Networks"
_MIDL.io/2021/Conference/Short — MIDL 2021 Poster_

### Official Review · Reviewer_8TBf · 2021-04-30

**Confidence:** 3
**Final Rating:** 2

**Summary:**

This paper is an abstract of the published journal paper: “Generating anthropomorphic phantoms using fully unsupervised deformable image registration with convolutional neural networks”. The authors present a registration-based method for generating realistic anthropomorphic phantoms. This is done by training an image registration network that uses the phantom 2d slice as the moving image and registers it to another CT slice.

**Strengths:**

- The paper tackles the important problem of generating realistic phantoms.
- The presented method shows better results than the other methods (SyN and Voxelmorph)
-	The authors introduce a “new” similarity measure by combining two established similarity measures.


**Weaknesses:**

-	The authors used some \vskip or any other format/spacing altering commands (see the spacing between abstract and introduction or any other new section) Without those commands the paper limit of 3 pages is not met.
-	The evaluation is only performed using SSIM and MSE and no further labeled data is used to validate the performance.
-	The weighting of the individual sections is not well chosen. For example, it is explained in the method section that the loss function is backpropagated and thus the parameters are updated or that a regularizer helps to obtain smooth deformation fields (Results and Conclusion).  At the same time, important information is missing to understand the method and results.
-	The deformation field looks highly irregular and seems to have quite a lot of foldings. This is not discussed at all.


**Deanonymize Review:**

no

**Detailed Comments:**

-	SSIM abbreviation is the abstract without explanation.
-	Table caption above the table not below.


**Justification Of The Rating:**

Even though the journal paper might present interesting results, the presentation in this short paper is not satisfactory. Especially because the authors hurt the 3-page limit, I argue to reject the paper.

**Paper Type:**

validation/application paper

**Special Issue:**

no

---

### Meta-Review · Area_Chair_N6h5 · 2021-05-09

**Recommendation:** Accept (Poster)
**Confidence:** 5

**Metareview:**

The paper contains interesting ideas and fits well within MIDL's short paper scope of also providing a live discussion space for accepted journal publications. Given that unfortunately only one reviewer was available, whose weak reject is mostly based on formatting, I believe that the paper can still be accepted. However, I strongly urge the authors to reformat their paper to be consistent with the MIDL style when preparing the camera-ready version.

---

### Decision · Program_Chairs · 2021-05-11

Accept (Poster)